# Development of Optical Sensors Based on Neutral Red Absorbance for Real-Time pH Measurements

**DOI:** 10.3390/s24175610

**Published:** 2024-08-29

**Authors:** Olaïtan Germaine Olorounto, Guy Deniau, Elisabeth Zekri, Denis Doizi, Johan Bertrand, Vincent Corbas

**Affiliations:** 1CEA-Saclay, DES/ISAS/DRMP/SPC/LC2R, F-91190 Gif-sur-Yvette Cedex, France; elisabeth.zekri@cea.fr (E.Z.); denis.doizi@cea.fr (D.D.); 2CEA-Saclay, DRF/IRAMIS/UMR-NIMBE/LICSEN, F-91190 Gif-sur-Yvette Cedex, France; guy.deniau@cea.fr; 3ANDRA, 1 Rue Jean Monnet, F-92290 Châtenay-Malabry, France; johan.bertrand@andra.fr (J.B.); vincent.corbas@andra.fr (V.C.)

**Keywords:** neutral red, absorption, diazonium aryl salt chemistry, electro-induced grafting, optical pH probe

## Abstract

Measuring pH with an optical sensor requires the immobilization of a chemical recognition phase on a solid surface. Neutral red (NR), an acid base indicator was used to develop two different optical probe configurations. The chemistry of aryl diazonium salts was chosen for the elaboration of this chemical phase, as it enables strong covalent bonds to be established on the surface of metallized glass or metallic surfaces. It also allows the formation of a thick film required to obtain an exploitable spectral response. The surfaces of interest (metallized optical fiber and 316 L stainless-steel mirror) are modelized by flat surfaces (metallized glass plates and 316 L stainless-steel plates). The analytical characterizations carried out (IR, XPS, UV-Visible, and profilometry) show that NR was covalently grafted onto the model surfaces as well as on the surfaces of interest. The supports grafted with NR to develop optical pH probes exhibit spectral changes, particularly the values of pKa, the pH range, and the isosbestic point wavelength. The experimental results show that the optical probe can be used for pH measurements between 4 and 8.

## 1. Introduction

The development and use of analytical tools for on-site measurements are of major importance in several applications, in particular, for environmental water quality monitoring in complex aqueous systems [1,2]. Among measurable quantities, a knowledge of pH value is fundamental because it governs a lot of chemical reactions that can induce important modifications in a complex aqueous system [3,4]. pH measurement is generally performed either by electrochemical (pH electrode) or optical (pH optode) methods [5,6,7,8]. Electrochemical methods are ideal for measurements in research laboratories but are limited for on-site measurements. They are fragile and require calibration with a reference electrode before measurement, and they suffer from acid and alkaline errors [9,10]. Optical methods offer advantages in terms of cost and simplicity of use; they do not require a reference electrode. They are not sensitive to electrical interferences and can address several measurement channels through a multiplexing configuration. They can be used for remote, continuous, real-time, in situ measurements and can operate in hostile environments [11].

The development of these optical probes is based on a simple concept involving the immobilization of a chemical recognition phase sensitive to pH variation on a surface area of the optical chain [12,13]. These methods use light to measure variations in optical properties resulting from the interaction between the aqueous system to be analyzed and the chemical recognition phase of the probe [14]. The immobilization technique for this chemical recognition phase is an important step in the development of these optical probes. The immobilized chemical layer will determine the nature of the chemical species to be measured, the optical characteristics of the probe, and its stability [15]. The literature lists three main methods for immobilizing colored pH indicators on the surface of a stable solid support that is permeable to the diffusion of hydronium and hydroxide ions: these are adsorption, entrapment, and covalent immobilizations.

Optical probes based on the adsorption of pH indicator molecules are easy and simple to manufacture. Their manufacture consists of dissolving the pH indicator molecule in a solution, then depositing it on the surface of interest. In this respect, Saari et Seitz [16] immobilized fluoresceinamine on cellulose, which is attached to the end of the bifurcated optical fiber. Safavi et al. [17] immobilized a mixture of Victoria blue and dipicrylamine on a triacetylcellulose membrane. Jeon et al. [18] developed a fiber-optic pH sensor based on a sol-gel film immobilized with neutral red. This method involves low-energy bonds between the chemical recognition phase and the surface of interest. Optical probes designed using this method suffer from limited practicability, as the recognition phase is slowly released into the analysis medium (as we verified ourselves using a commercial optode).

Optical probes are based on the use of pH indicator molecules trapped in a polymer matrix. This method involves trapping the pH indicator molecule in a polymer matrix. Egami et al. [19] developed a fiber-optic pH probe based on the evanescent absorption of methyl red immobilized in poly(methyl methacrylate) fiber cladding. Goicoechea et al. [20] fabricated evanescent wave optical-fiber pH probes using layer-by-layer (LBL) electrostatic self-assembly methods. This LBL technique involves the utilization of neutral red as a pH indicator and poly(allylamine hydrochloride) (PAH) and poly(acrylic acid) (PAA) as polycation and polyanion products, respectively. Kirkbright et al. [21] constructed an absorbance-based pH probe that exploits the spectral changes in bromothymol blue as a pH indicator. Bromothymol blue was adsorbed on styrene/divinylbenzene copolymers. This method requires the pH indicator molecule to be soluble and homogeneously distributed in the polymer matrix, without forming aggregates. Methods based on adsorption and trapping lead to the poor robustness of the probe because the pH-sensitive phase undergoes a temporal degradation of its response [22].

Other optical pH probes are based on the adsorption of indicator molecules on a Bragg grating, such as those described by Liang et al. [23], Aldaba et al. [24], and Janting et al. [25]. The optical pH probes developed using this method do not solve the stability problem, as the hydrogel is just deposited on the Bragg grating.

Other authors also describe pH sensors based on the principle of surface plasmon resonance, such as Antohe et al. [26] and Singh et al. [27].

All these different methods use the physisorption of the active principle, and therefore do not create a sensor that is stable over time.

Covalent immobilization reduces the problems associated with the degradation of the chemical recognition phase. It can be achieved by either chemical or electrochemical grafting. The literature lists countless optical pH probe publications using chemical grafting [28,29,30,31,32,33,34,35,36,37,38]. The design process for these pH sensors based on chemical grafting requires a substrate surface activation step by using various chemical methods (silanization, epoxidation, urea, and formaldehyde). The chemical recognition phase was simply realized by immersing the surface of the activated substrate in a solution containing a colored pH indicator having a specific function (e.g., an aromatic primary amine function) capable of reacting with the surface to bind covalently. The development of this chemical recognition phase by the chemical grafting process is much slower and results from a delicate balance between activated substrate/pH indicator interactions. This chemical grafting process leads to the formation of a monolayer.

However, it is important to immobilize a sufficiently thick organic layer to obtain an optical pH sensor that delivers a valuable spectral response. Electrochemical grafting involves the establishment of a covalent bond between the substrate surface and the chemical recognition phase, and generally leads to the formation of a thick organic layer [39,40,41,42,43,44]. The thickness of the electro-grafted organic layer can reach several hundred nanometers [45]. The thick film formed is stable and resistant to many solvents and even in the presence of ultrasounds [46].

Redox grafting of RN onto a surface using the Graftfast^TM^ process has been reported in the literature [47], but this process is not reproducible and does not allow the control of the thickness of the chemical recognition phase.

To the best of our knowledge, no study was reported on the electrochemical grafting of NR by the cathodic reduction of aryl diazonium salts to realize an optical pH probe. The chemistry of diazonium aryl salts was used to create covalent bonds between the chemical recognition phase and our conductive surfaces. The major advantages of this electrochemical method under surface polarization are its ability to quickly generate radicals in a gentle and progressive manner, observed on recorded voltammograms [48]. The thickness of the grafted chemical recognition phase was controlled through the concentration of diazonium salts used, the amount of charge consumed during the electro-grafting process, the number of grafting cycles, and the scanning speed. The chemical composition of the recognition phase is controlled by the various analytical characterizations carried out, particularly X-ray photoelectron spectroscopy [39].

The aim of this work is to covalently immobilize NR on the end of a metallized optical fiber and on a 316 L stainless-steel mirror, parts of the optical chain, in order to choose the best location for the chemical recognition phase. The methodology for forming the chemical recognition phase involves a combination of electro-induced grafting and covalent immobilization, as described in the literature [49]. For this purpose, the metallized optical fiber and the 316 L stainless-steel mirror were modelized by a metallized glass plate and a 316 L stainless-steel plate, respectively, in order to optimize the electro-grafting process and make analytical characterizations easier. The chemical recognition phase of electro-grafted NR is characterized by profilometry to determine its thickness, X-ray photoelectron spectroscopy (XPS) to determine its chemical composition, infrared spectroscopy to identify its molecular structure, and UV-Visible spectroscopy to measure its absorbance. This optical pH probe can find applications for continuous pH measurements and especially for remote measurements in complex aqueous systems. This is the case, for example, for the future Cigeo project, which is a deep geological disposal facility for radioactive waste to be built in France [50]. The French Geological Disposal will be managed by Andra (the French national radioactive waste management agency). The practical application of this optical pH probe in complex aqueous environments, such as those anticipated in deep geological disposal facilities, helps avoid chemical changes, such as carbon dioxide partial pressure. pH variation in the pore water of geological formations is correlated with the existence of chemical reactions involved in the degradation of clay, metals, and concrete.

## 2. Experimental Section

### 2.1. Reagents

All solutions were prepared with ultra-pure Milli-Q water (18.2 MΩcm). Neutral red and sodium nitrite (Sigma Aldrich Chimie, Saint-Quentin-Fallavier, France) buffer solutions of different pH levels (Fisher Scientific SAS, Illkirch, France) were used without purification.

### 2.2. Optical Sensor Experimental Setup

Two different optical configurations are studied in this work (Figure 1a,b). In the first configuration, the chemical recognition phase was immobilized on the end of a two-fiber octopus. Each optical fiber had a core diameter of 940 µm and a numerical aperture of 0.22 (Figure 1c). In the second optical configuration, the recognition phase was immobilized on the 316 L stainless-steel mirror. Both optical configurations required a mirror to reflect light rays back to the receiving fiber for analysis by the detector. The geometry of the 316 L stainless-steel mirror had a diameter of 12.7 mm and a radius of curvature of 24 mm. The mirror holder was used to integrate the mirror with the two-fiber octopus (Figure 1d). This mechanical support is designed to match the mirror’s geometry. For optimum convergence of the light rays emerging from the mirror, the distance between the mirror and the receiving fiber input must be equal to the value of the radius of curvature (24 mm). This means that the mirror holder also had a length of 24 mm.

About the operating principle of the optical pH sensor, a white-light source was injected into an emission optical fiber. This fiber carried the light to the measurement location. A mirror positioned on front of the fiber sent the light back to a receiving optical fiber to a spectrometer for spectral analysis. Light rays from the source emerged telecentrically from the output diopter of the emissions fiber (Figure 1e). Computer software enabled spectral response processing. The interaction of the immobilized organic layer with the protons contained in a given sample, in the presence of light, led to variations in optical properties.

The light source (HL 2000-FHSA), the optical fibers, and the spectrometer (HR 2000) were supplied by Ocean Insight. The mirror and the mechanical support were manufactured by OPA OPTICAD(Le Plessis-Belleville, France). The mirror holder was designed using SolidWorks software (*SolidWorks software 2021*) and manufactured by Faulcon, Saclay, a company specialized in precision mechanics. The connections for the optical fibers were of the SMA type and the mirror was mechanically fixed in position using a 6-sided screw.

### 2.3. Substrate Preparation

#### 2.3.1. Glass Plates

The optical fiber was modelized by a glass plate. The glass plates were metallized to provide the conductive surface required for electrochemical grafting using aryl diazonium salt chemistry. The thickness of the metal deposit was low (30 nm) to allow UV-visible transmission measurements. The glass plates were first cleaned in an ultrasonic tank for 5 min in three different solvents: acetone, ethanol, and water. After cleaning, the glass plates were dried with nitrogen to remove any traces of water. The glass plates were placed in an evaporation vacuum chamber (10^−7^ mbar). Both 5 nm chromium and 25 nm gold were deposited on the plates. These metallized glass plates were used without further treatment.

#### 2.3.2. Stainless-Steel Plates

The 316 L stainless-steel mirror was modelized with a 316 L stainless-steel plate. It is better to perform optimization studies on these 316 L stainless-steel plates because of the cost of the mirror and for an easier covalent grafting process and analytical characterizations. The 316 L stainless-steel plate was first polished with different papers containing Struers silicon carbide (SiC). The various SiC-type papers (P80 to P4000) and grain-size papers (200 to 5 µm) were used to obtain identical and optically reflective surfaces. After polishing, the plate was treated with diamond paste (¼ µm), then rinsed with ultra-pure water and dried with argon. Then, the plate was ultrasonically cleaned in a mixture of water (5 mL), acetone (5 mL), and ethanol (5 mL) for 10 min, and then in acetone only for 10 min before being used.

### 2.4. Preparation of the Reaction Medium

NR grafting was carried out in HCl (0.1 M) acidic aqueous medium. An equimolar quantity of NR and sodium nitrite was used to obtain the diazonium salt of NR. For each concentration, NR was dissolved in acidic HCl (59 mL) and the solution was stirred for 5 min. Sodium nitrite was first dissolved in ultra-pure water (1 mL) and then added to the NR solution to transform the aromatic amine into cation diazonium. The reaction was complete with a 100% yield. The mixture, stirred for 5 min, was then transferred into a three-electrode electrochemical cell.

### 2.5. Electrochemical Measurements

All experimental tests were performed in a three-electrode electrochemical cell at room temperature using a Biologic SP-300 potentiostat controlled by EC-Lab software (v11.61). The reference electrode was a silver wire (pseudo-reference), the working electrode was a metallized glass plate (5 nm Cr/25 nm Au) held in place with alligator clips, and the counter-electrode was a platinum wire (for the cases of the metallized glass plate and metallized optical fiber). For all experiments carried out on 316 L stainless-steel plates and on the 316 L stainless-steel mirror, the counter-electrode used was a carbon plate. All voltammograms were recorded at 5 mV/s (for the cases of the metallized glass plate and metallized optical fiber) with an inversion potential of −1 V. For all experiments carried out on the 316 L stainless-steel plates and the 316 L stainless-steel mirror, the scanning speed used was 50 mV/s. The immerged surfaces were ~cm^2^ for the metallized glass plates, the stainless-steel plates, and the mirror.

### 2.6. Spectroscopic Measurements

All infrared measurements were performed with a Bruker vertex 70 spectrophotometer controlled by OPUS acquisition software (v7.8). This FTIR spectrophotometer is equipped with an ATR Pike accessory, enabling measurements in the attenuated total reflection (ATR) mode with a DTGS detector. FTIR spectra were recorded in the 600 to 4000 cm^−1^ wavenumber range with a spectral resolution of 4 cm^−1^ using 16 scans.

All XPS spectra were recorded using Thermo Scientific’s ESCALAB 260Xi X-ray photoelectron spectrometer. This spectrometer offers high sensitivity and resolution; it uses a 1486.6 eV monochromatic Al K-Alpha X-ray source capable of analyzing a 900 µm diameter area of a sample to a depth of 10 nm.

All UV-Visible absorption spectra were recorded in a Hellma 10 mm quartz cell using a Cary 60 spectrophotometer (Agilent technologies, Les Ulis, France) controlled by Cary WinUV software (*version 5*). Absorption spectra were recorded at a speed of 4800 nm/min.

## 3. Results and Discussion

### 3.1. Study of NR in Solution by UV-Visible Spectrophotometry

Prior to any electro-induced covalent functionalization, NR was studied in solution. Its pKa value is 7.1 in an acidic medium; it turns yellow in a basic medium (Figure 2).

Figure 3 shows the UV-Visible absorption spectra of NR recorded in buffer solutions of different pH levels. These spectra show two absorption bands corresponding to its acidic and basic forms. The two bands intersect at a point called the isobestic point (470 nm). This point indicates the presence of an equilibrium between the two forms of NR. The curve of absorbance versus pH measured at 550 nm has a sigmoidal shape. From this curve, we can deduce that NR shows linear absorbance variation as a function of pH for pH values between 6 and 8 (Figure 4).

### 3.2. Electro-Induced Grafting of the Diazonium Salt of RN onto a Metallized Glass Plate

The diazonium salt of RN was grafted by a cathodic reduction onto a metallized glass plate. The electro-grafting reaction is carried out by cyclic voltammetry and sequencing. After each electro-grafting cycle, a plate washing protocol is performed. The substrate is periodically rinsed with water, then ultrasonically treated in a mixture of water (5 mL) and ethanol (5 mL). This washing protocol between each voltammetric cycle makes the film thicker while progressively eliminating the physisorbed part of the indicator. This avoids building up successive layers on the physisorbed parts of the coating under construction. The voltammograms recorded during the electro-reduction of NR are shown in Figure 5. An NR reduction peak was observed at around—0.2 V/Ag and gradually disappeared.

### 3.3. Optimizing the Initial Concentration of the Diazonium Salt of NR on a Metallized Glass Plate

The sensitivity of the pH probe depends significantly on the total amount of indicator grafted onto the metallized substrate. It is therefore important to determine the optimum concentration for effective covalent grafting to obtain a thick film. Five different metallized plates were functionalized at different concentrations of NR: 2.5; 5; 7; 10, and 14 mM. These metallized and grafted glass plates were characterized by UV-Visible, IR-ATR, and profilometry.

Figure 6 shows the UV-Visible characterizations carried out in buffer solutions ranging from pH 1 to pH 10. In this figure, the optical response of NR measured was related to the total quantity immobilized on the metallized glass plate. Thus, the absorbance of grafted NR increases as a function of the initial concentration up to a limiting value of 10 mM. The value of 10 mM can therefore be considered as the optimum concentration. However, at higher concentrations, such as 14 mM, too many radicals were generated in the reaction medium. The radical polymerization reaction takes place in solution rather than on the surface. The radicals that were formed dimerized in solution. The kinetics of this reaction varies quadratically with the concentration of these radicals. As a result, as the initial NR concentration increases, reactions in solution are favored, increasing the physisorbed portion of the coating. This results in the loss of radicals available for grafting. Dimerization products physisorbed on the surface were removed during the washing protocol. This may explain the lower quantity of 14 mM NR grafted on the metallized glass plate.

After grafting, the pH range of NR is between pH 4 and pH 6, whereas in solution it is between pH 6 and pH 8. Its pH range was therefore shifted toward an acidic pH. This is due to the stacking of NR units to form pyrazines on the metallized surface and the loss of the amine function during electrochemical reduction. The loss of the amine function induces a change in the aromatic conjugate system responsible for adsorption in the visible neutral red. After grafting, a multilayer is formed on the metallized surface, whose mechanism is described in the literature [49]. Interestingly, the pH range of grafted NR remains constant, whatever its initial concentration in solution.

Figure 7 shows the UV-Visible absorption spectra of NR grafted on a glass plate metallized at 10 mM. This figure presents two absorption bands (acid and basic forms) similar to the ones observed in solution. Comparing this result with the one obtained for the neutral red in solution, we can observe that there are remarkable changes in the optical properties, particularly at the isobestic point for the grafted molecule (500 nm). Grafting resulted in a pH-sensitive range shift, as mentioned above. The grafting of NR therefore led to a modification of its optical characteristics (pKa, pH range, and isobestic point).

The thickness of metallized NR-grafted glass plates was determined by profilometry. The results obtained are illustrated in Table 1. According to this study, the metallized glass plate grafted with 10 mM of NR is the thickest.

The reproducibility of NR grafting onto metallized glass plates is studied. Three different metallized glass plates from the same batch were grafted with 10 mM of NR (Figure 8). In order to compare the results, absorbances measured at 550 nm for each grafted glass plate were normalized by the absorbance obtained at pH 2 for each grafted glass plate. The results obtained show that the spectral response of these three grafted glass plates is reproducible.

A cycle of deprotonation and protonation was performed to check for possible hysteresis at 10 mM (Figure 9). The results obtained on the forward and return paths are almost identical. We can therefore conclude that the spectral response of grafted NR is reversible.

The metallized glass plate grafted with 10 mM of NR was characterized by IR-ATR. On this grafted glass plate, 10 different locations were analyzed (Figure 10). The IR-ATR spectra obtained at these different points are almost identical. We can therefore conclude that the grafted NR organic layer is homogeneous.

### 3.4. Transposition of NR Grafting on a Metallized Optical Fiber

The NR grafting process previously optimized on glass plates was transposed to the end of a metallized optical fiber (diameter: 940 µm) (Figure 11). After grafting, the fiber end exhibits a red color visible to the naked eye. This confirms that grafting has indeed taken place. This NR-grafted optical fiber was characterized in a simple experimental setup, shown in Figure 1a.

Figure 12 shows the absorption spectra of NR grafted onto the optical fiber as a function of pH. The spectra show a broad absorption band between 500 and 550 nm. The absorption spectra of NR grafted on the optical fiber have a lower sensitivity compared to those measured on the metallized glass plates. The electro-grafted organic film on the optical fiber is therefore thinner.

The absorption spectra data processing at 550 nm makes it possible to plot the absorption variation curve as a function of pH (Figure 13). The shape of the experimental curve can be modelized with a line of equation y = (−0.1)x + 1.27, where x is the pH value and y is the optical density measured in the sample. This line has a regression coefficient (0.98) and a slope of −0.1 per pH unit. It is important to note that NR grafted on the metallized optical fiber has a pH range between 4 and 8, while that obtained on the metallized and grafted glass plate varies between 4 and 6. The optical characteristics of grafted NR may therefore depend on the geometry of the electrochemical assembly used.

It is also interesting to explore the electro-induced covalent grafting of the same molecule on the 316 L stainless-steel mirror. From a practical point of view, grafting onto the mirror’s surface avoids the metallization step in the case of fiber. The control of the layer thickness and the chemical composition is the same for the two types of surfaces (316 L stainless-steel mirror and metallized fiber).

### 3.5. Electro-Induced Grafting of NR on a 316 L Stainless-Steel Mirror

Grafting NR on a 316 L stainless-steel mirror was also evaluated (Figure 1b). It was chosen because of its corrosion resistance in the presence of a strong acid medium (condition of grafting NR). The optical design of the stainless-steel mirror requires the optimization of its geometry (diameter and radius of curvature).

The mirror’s geometry was optimized using Zemax/optic studio software (version 20.1). The software was used in the non-sequential mode. The geometric image analysis feature was used to analyze the quality of the light spot at the input of the receiving fiber. This feature calculates the coupling efficiency by taking into account the numerical aperture of the optical fibers. This coupling efficiency corresponds to the ratio between the quantity of light rays at the input of the receiving fiber and the quantity of rays at the output of the transmitting fiber. This software, based on the tracing of light rays, allows us to visualize the light spot after reflection on the mirror. Thanks to this software, different configurations can be studied by varying the diameter of the mirror and its radius of curvature in order to compare the results and to choose the best result (Table 2). When the mirror’s geometry is optimal, the light spot after reflection is as homogeneous as possible. Based on these simulations, we chose to order stainless-steel mirrors with a radius of curvature of 24 mm and a diameter of 12.7 mm. The diameter was chosen so that the beam was not diaphragmed. Increasing the radius of curvature increases the coupling efficiency by limiting optical aberrations, but makes the system bulkier and more sensitive to misalignments. The chosen radius is the result of a compromise.

The grafting of NR was carried out by cyclic voltammetry and by sequencing on the 316 L stainless-steel plate. At the end of each cycle, the stainless-steel plate was removed, and then rinsed with water and subjected to ultrasound for one minute in a mixture of water (5 mL) and ethanol (5 mL). The grafting cycles were recorded by sequencing on the stainless-steel plate at 50 mV/s.

The analysis of the recorded voltammograms shows that the electro-reduction of NR occurred at around—0.1 V/Ag (Figure 14). The evolution of the observed current leads to the formation of a multilayer, and thus to the passivation of the stainless-steel surface. The observed decrease in the current was related to the formation of an insulating film that passivated the surface of stainless steel.

### 3.6. Optimizing the Initial Concentration of the Diazonium Salt of NR on 316 L Stainless-Steel Plates

Obtaining a thick organic structure grafted on the surface of a substrate requires the optimization of electrochemical conditions, in particular, the initial concentration of diazonium salt. Different concentrations of 5, 10, and 14 mM were studied in 0.1 M of HCl aqueous medium on the 316 L stainless-steel substrate.

Different 316 L stainless-steel plates grafted with NR at variable concentrations were characterized in XPS (Figure 15). In the three samples, the intensity of the C1s and N1s peaks increases after the functionalization of the surface of the stainless-steel metal substrate. This indicates the presence of organic film on the surface of the stainless-steel metal substrate.

For the first two samples grafted with NR (5 and 10 mM), the signal of the metal substrate, in particular, that of Cr, is visible, which is not the case at 14 mM. XPS allows the analysis of only about the first 10 nanometers of the sample. This characterization confirms the presence of a thick organic layer of NR on the surface of the stainless steel with a thickness greater than 10 nm for the concentration of 14 mM. The film grafted with NR at 14 mM therefore appears thicker. At this concentration, the thickness estimated by profilometry is close to 100 nm.

The covalent grafting of NR at 14 mM was therefore transposed onto the 316 L stainless-steel mirror. The fluorine signal observed on the XPS spectra appears to be related to sample contamination.

### 3.7. Transposition of NR Grafting on 316 L Stainless-Steel Mirror

The electro-induced grafting procedure of NR studied at 14 mM on 316 L stainless-steel plates was transposed onto the 316 L stainless-steel mirror. Figure 16 shows two stainless-steel mirrors, one grafted and one not. At the end of grafting, a red color is clearly visible on the mirror. This indicates that grafting has taken place. We also notice that the organic layer of grafted NR is homogeneous on the mirror.

A two-channel optical configuration (one reference channel and one sample channel) was explored to evaluate the performance of the organic layer of the grafted NR. This configuration uses a single light source (HL-2000-FHSA), a 2 × 2 FOS TTL multiplexer, a spectrometer (Flame 3648 pixels model), 600 µm diameter optical fibers, two 316 L stainless-steel mirrors (one not grafted and one NR grafted), and a computer with SpectraSuite acquisition software (version 2.0.0).

The UV-Visible absorption spectra recorded in different pH buffer solutions were presented (Figure 17). These spectra show a broad absorption band at 450 and 550 nm. The two acid-base forms are difficult to detect. The spectral response of NR obtained is selective but not sufficiently sensitive in the different buffer solutions of different pH values. We believe these results are due to the fact that the organic layer of the grafted NR is not thick enough to deliver a valuable spectral response. These spectra were analyzed at 550 nm in order to plot the absorption variation curve as a function of pH (Figure 18). This curve is modelized with a trend line of equation: y = (−0.13)x + 1.7. The regression coefficient of this line is 0.97 with a slope of (−0.13) per pH unit.

## 4. Conclusions

The experimental work carried out on the electro-induced covalent grafting of NR in an acid medium (0.1 M) confirms that a homogeneous organic layer, reversible, and reproducible with a thickness between 100 and 300 nm is grafted on the surface of metallic and metallized substrates (316 L stainless steel and gold, respectively). This covalent grafting process can be transposed onto a metallized optical fiber and a 316 L stainless-steel mirror. The optical properties of RN obtained in solutions are different from those obtained on surfaces. The spectral response of NR is selective and sensitive in the low acid pH range, but its sensitivity must be improved. These experimental results prove that this acid-base pH indicator can be used as a chemical recognition phase to successfully develop two optical sensor configurations. We think that the covalent functionalization of the surface of the 316 L stainless-steel mirror is much more relevant than that of the metallized optical fiber because the mirror has a larger diameter than that of the optical fiber. Furthermore, the metallized fiber has lower light transmission and is less practical when performing measurements. It is therefore possible to immobilize larger quantities of NR molecules on the mirror’s surface and thus create a sufficient organic layer to deliver an exploitable spectral response. This experimental work proves that covalent conductive surface functionalization by the cathodic reduction of diazonium aryl salts represents a rapid approach and an alternative to sol/gel functionalization methods in order to develop a robust optical pH probe.

The two configurations of the optical probe were tested in the Underground Laboratory in Bure, France. The initial results are very promising.

## Figures and Tables

**Figure 1 sensors-24-05610-f001:**
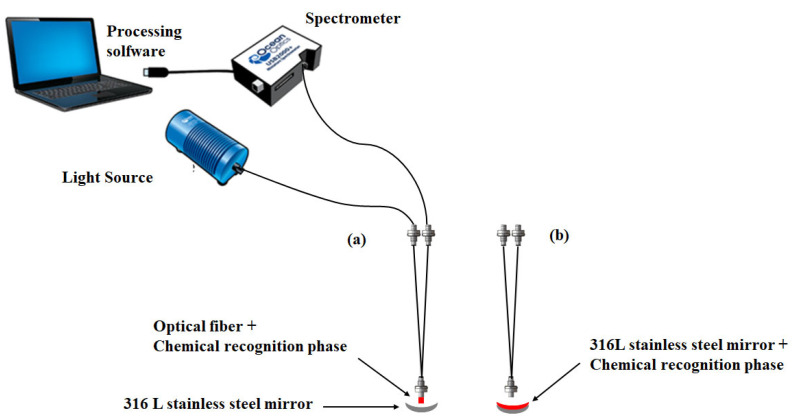
Schematic diagram of optical pH probe: immobilization of chemical recognition phase on the optical fiber (**a**) or on the stainless-steel mirror (**b**). Microscope image of the fiber tip (**c**) and integration of the mirror with the optical fibers (**d**). Optical diagram (**e**).

**Figure 2 sensors-24-05610-f002:**
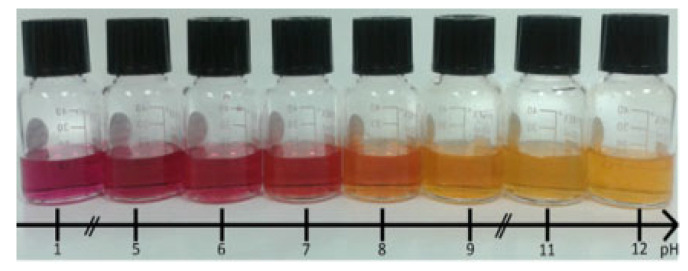
Evolution of NR color with pH in different buffer solutions.

**Figure 3 sensors-24-05610-f003:**
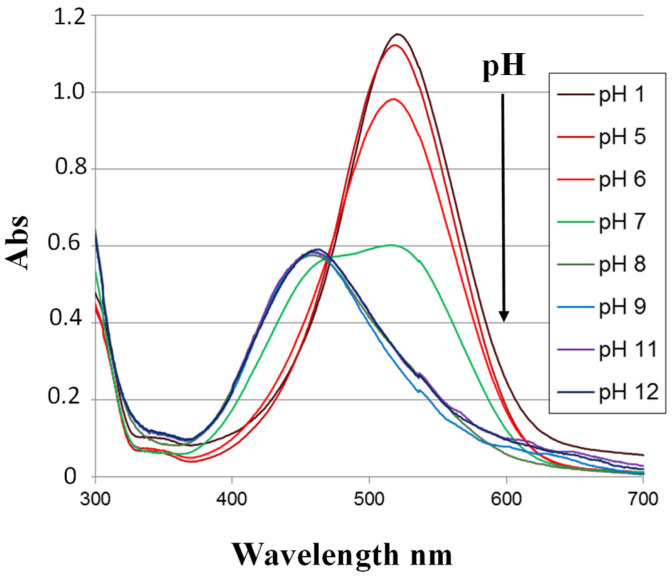
UV-Visible absorption spectra of NR in solution as a function of pH.

**Figure 4 sensors-24-05610-f004:**
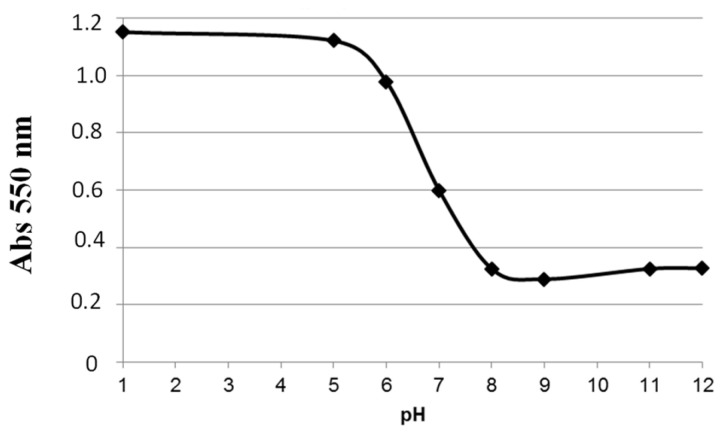
NR absorbance versus pH curve at 550 nm.

**Figure 5 sensors-24-05610-f005:**
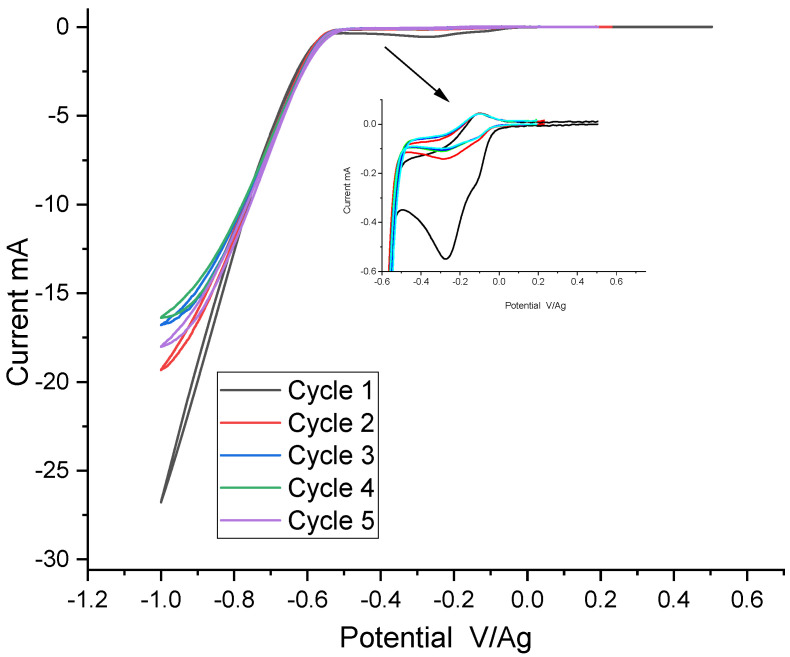
Voltammograms recorded during the electro-reduction of NR on a metallized glass plate: 5 cycles, in HCl 0.1, RN 10 mM, and NaNO_2_ 10 mM.

**Figure 6 sensors-24-05610-f006:**
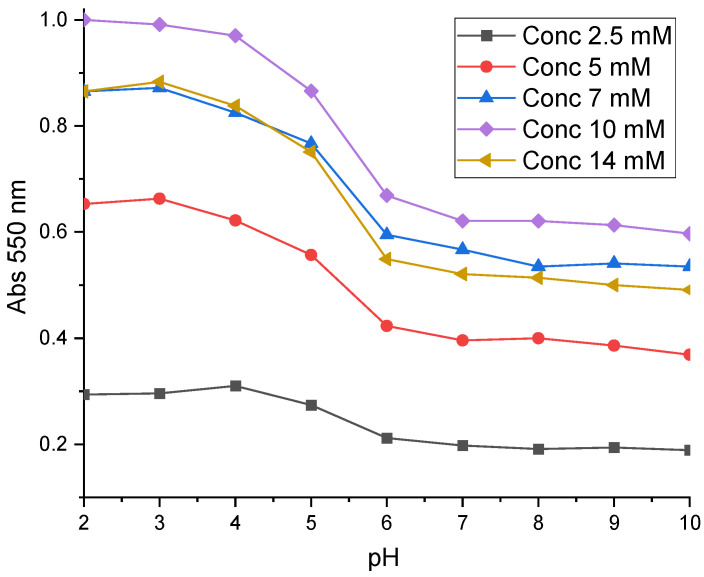
UV-Visible characterization of five different RN-grafted metallized glass plates as a function of pH.

**Figure 7 sensors-24-05610-f007:**
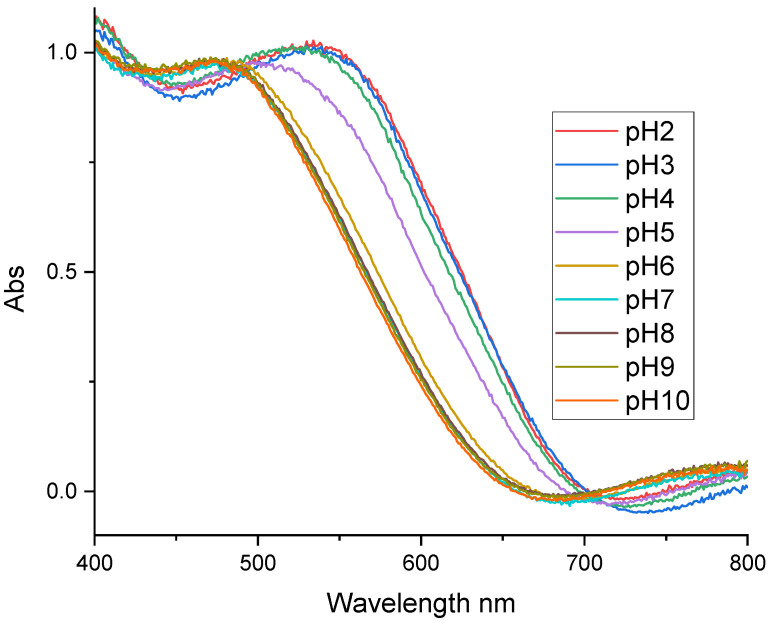
Absorption spectra of the NR film grafted onto the metallized plate at 10 mM and in buffer solutions of different pH levels.

**Figure 8 sensors-24-05610-f008:**
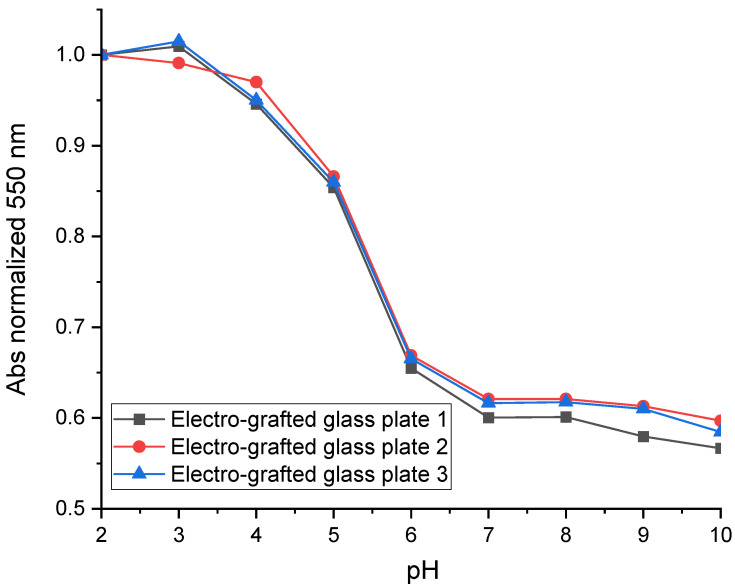
Reproducibility on three different metallized glass plates grafted with NR.

**Figure 9 sensors-24-05610-f009:**
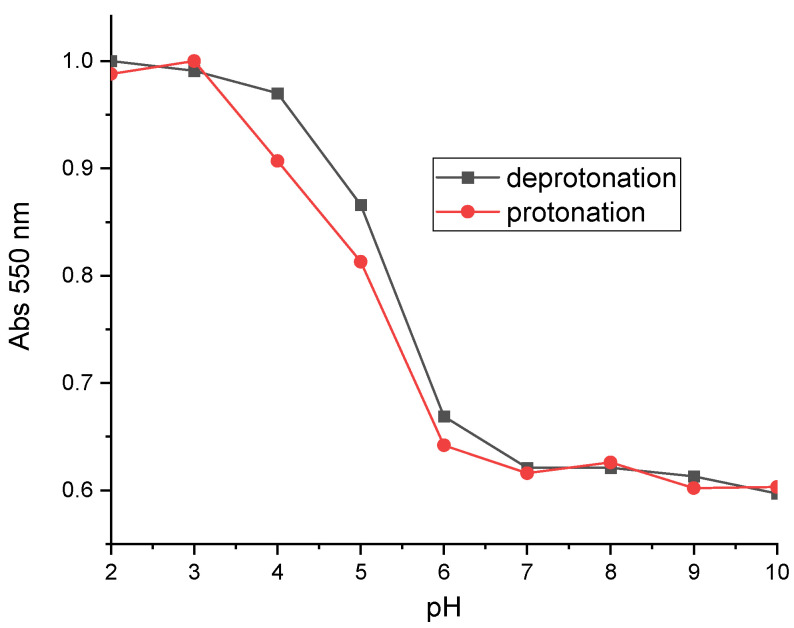
Deprotonation and protonation cycles performed on a metallized glass plate grafted with 10 mM of NR.

**Figure 10 sensors-24-05610-f010:**
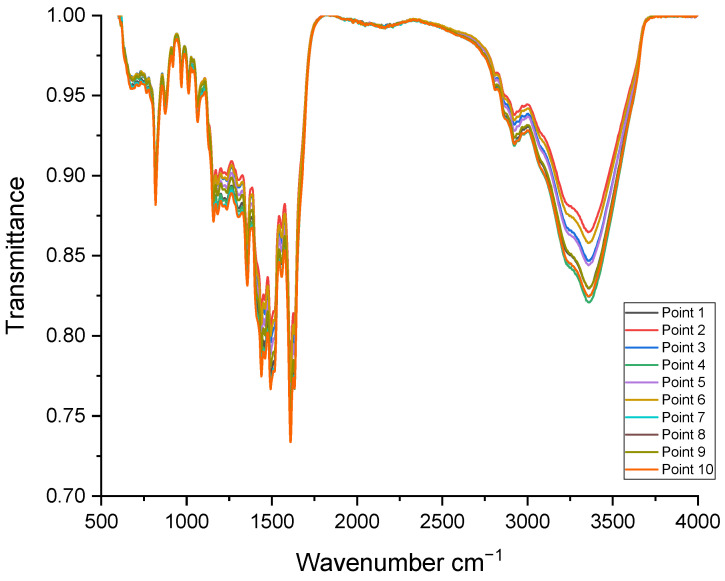
IR-ATR characterization of the metallized glass plate grafted with 10 mM of NR.

**Figure 11 sensors-24-05610-f011:**
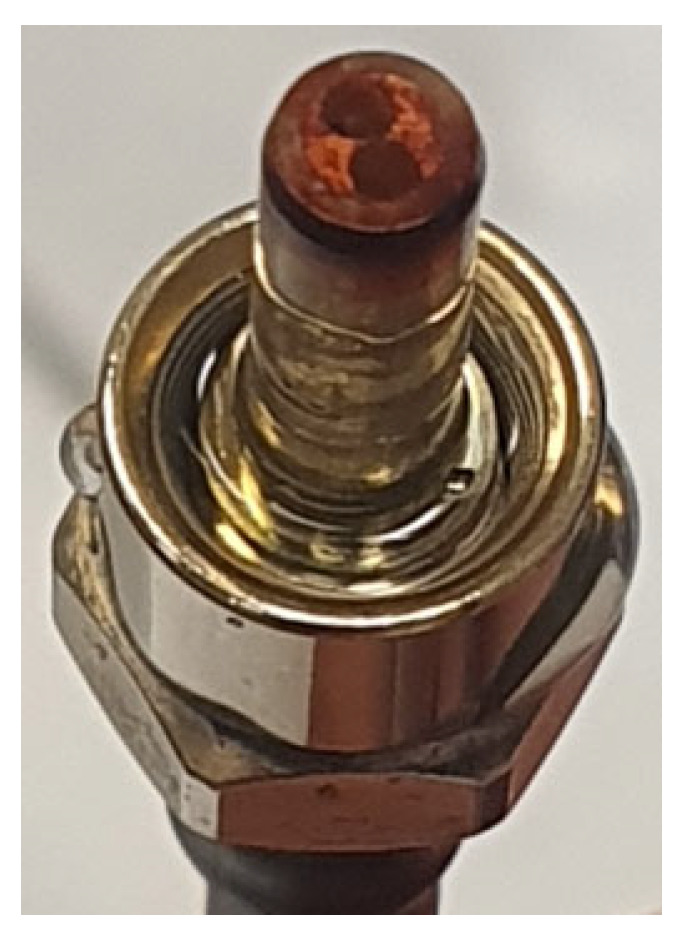
Grafting of NR on the end of a two-way metallized optical fiber.

**Figure 12 sensors-24-05610-f012:**
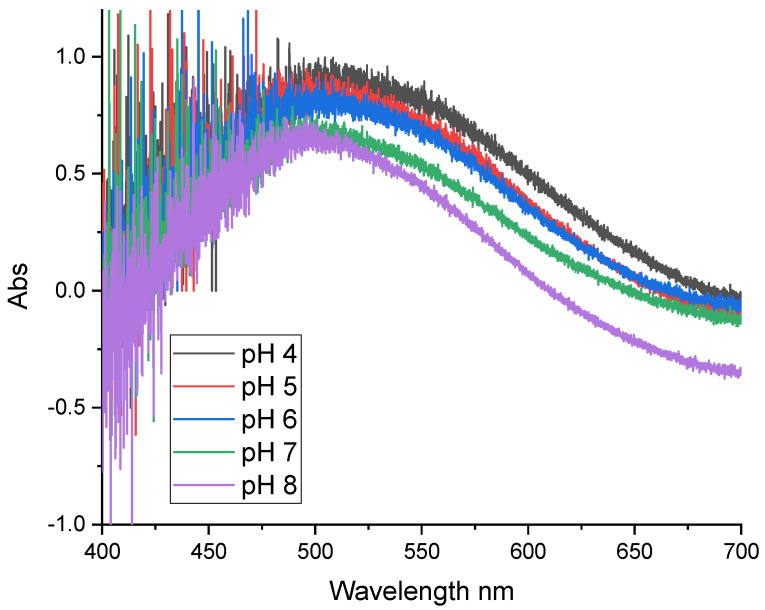
UV-Visible absorption spectra of NR grafted on the optical fiber in solutions of different pH values.

**Figure 13 sensors-24-05610-f013:**
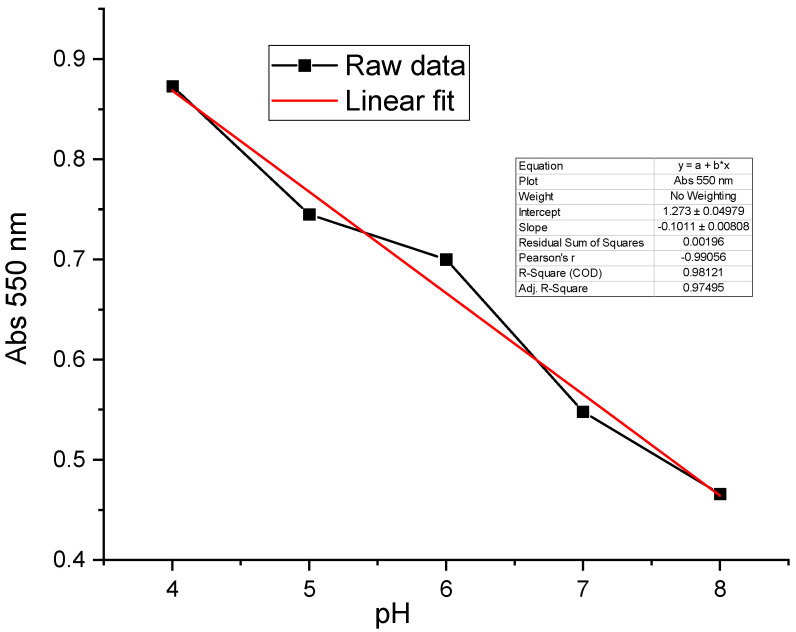
Absorbance versus pH curves for NR grafted on the optical fiber.

**Figure 14 sensors-24-05610-f014:**
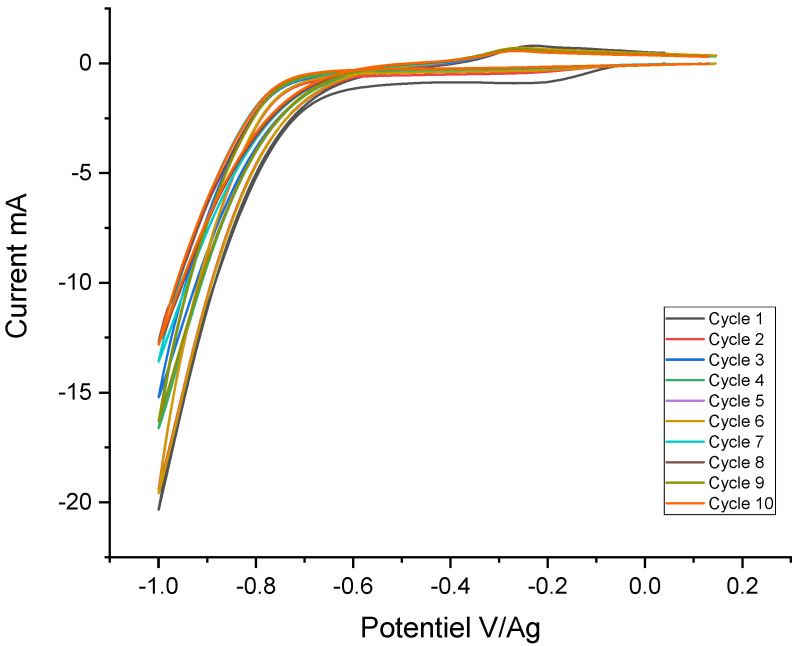
Voltammograms recorded during electro-grafting of NR 5 mM and 5 mM NaNO_2_.

**Figure 15 sensors-24-05610-f015:**
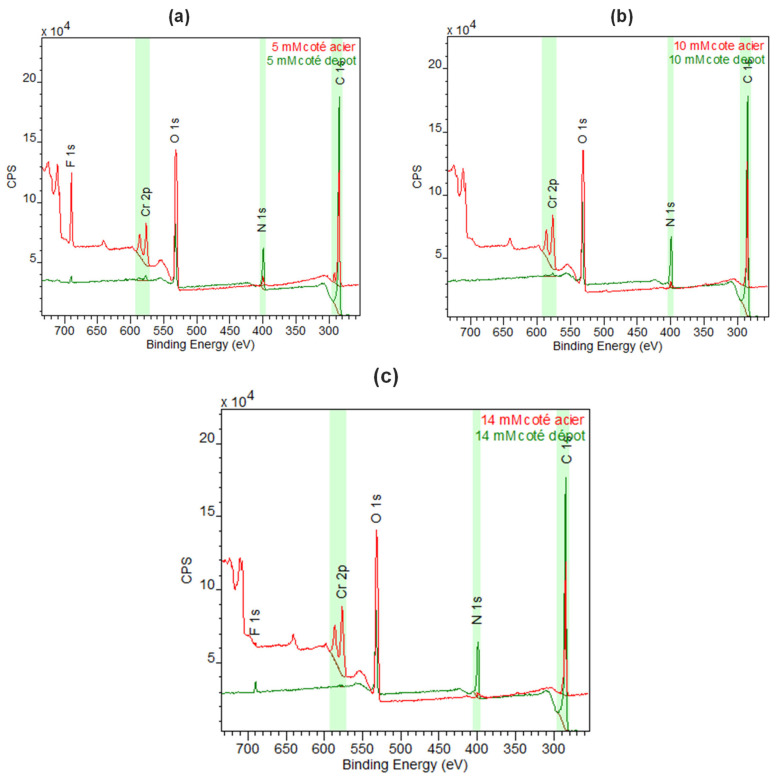
Overall surface analysis obtained using XPS on the stainless-steel plates grafted with NR: (**a**) = 5 mM, (**b**) = 10 mM, and (**c**) = 14 mM.

**Figure 16 sensors-24-05610-f016:**
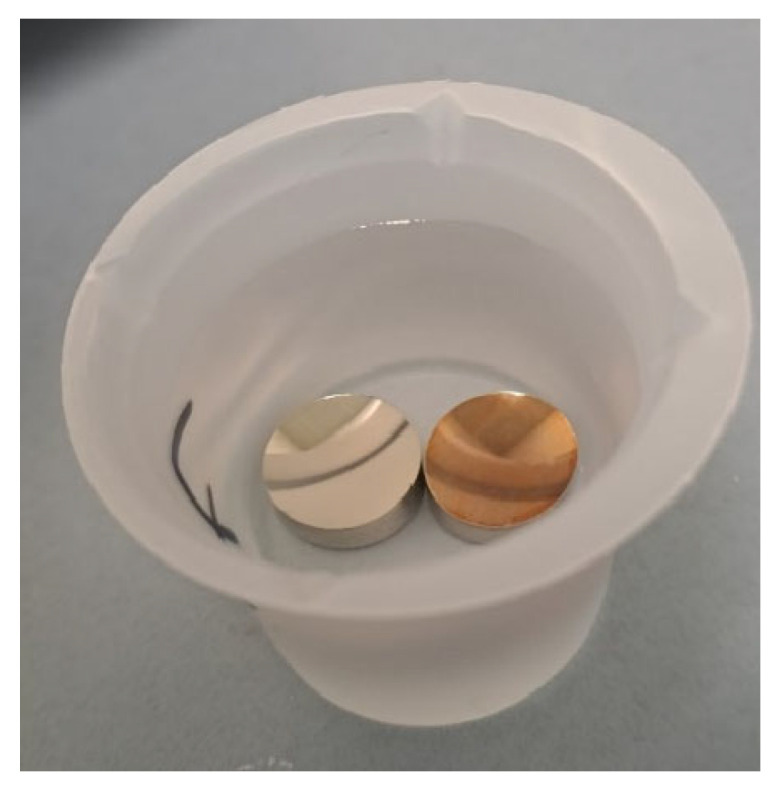
Two stainless-steel mirrors, one grafted and the other not grafted.

**Figure 17 sensors-24-05610-f017:**
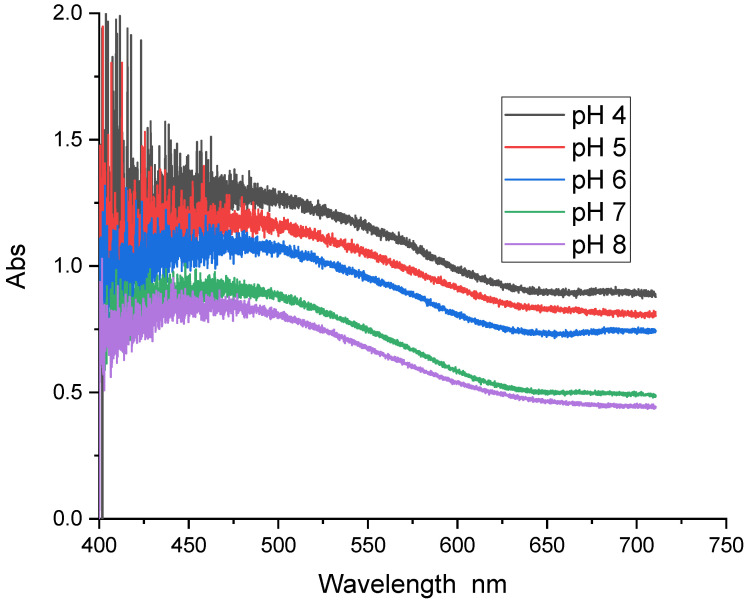
UV-Visible spectra of NR grafted on a 316 L stainless-steel mirror in buffer solutions of different pH values; optical configuration with two measurement channels and a light source.

**Figure 18 sensors-24-05610-f018:**
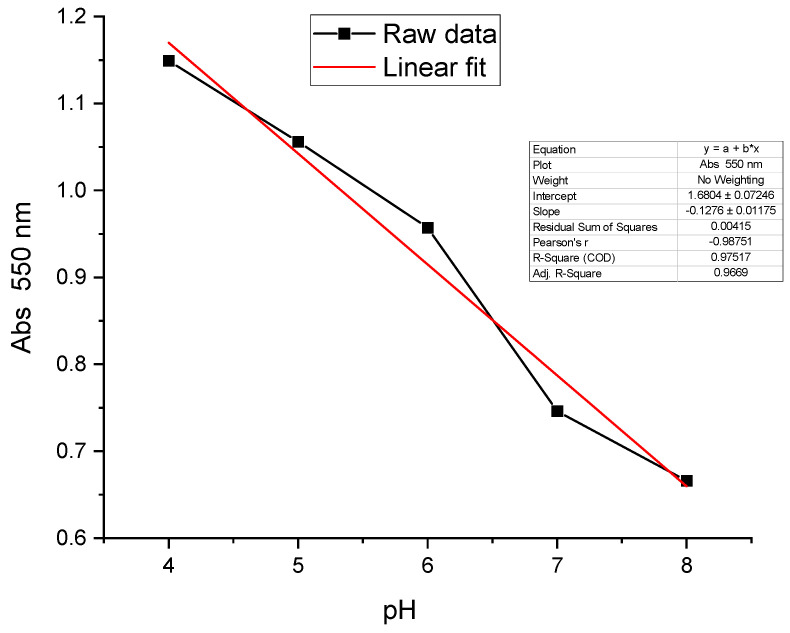
Absorption variations at 550 nm versus pH of NR grafted on the 316 L stainless-steel mirror.

**Table 1 sensors-24-05610-t001:** Estimated thicknesses as a function of concentration.

Concentration (mM)	Thickness (nm)
2.5	150
5	200
7	250
10	300
14	250

**Table 2 sensors-24-05610-t002:** Optimized mirror geometry in 316 L stainless steel, using Zemax/optic studio software (version 20.1).

Mirror Geometry		Quality of the Input Light Signal	
Diameter (mm)	Radius of curvature (mm)	Coupling efficiency (%)	Light spot
5	4	38	Not homogeneous
	10	67	Not homogeneous
12.7	19	82	Homogeneous
	24	86	Homogeneous

## Data Availability

Data are contained within the article.

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
