# Peer review of "Development of Optical Sensors Based on Neutral Red Absorbance for Real-Time pH Measurements"

_sensors, 2024, doi:10.3390/s24175610_

Round 1

Reviewer 1 Report

Comments and Suggestions for Authors

This paper presents an optical pH sensor that utilizes immobilized Neutral Red on metalized surfaces. The authors stated that the sensor demonstrates strong covalent bonding, facilitating the formation of a thick film essential for spectral response. The experimental analyses showed the sensor's capability to measure pH effectively within the mildly acidic range (pH 4-8). The design approach is fine. Several concerns require attention in the revision process.

1.     To provide a thorough analysis of micro pH sensors, authors should expand their comparison to encompass a broader range of sensing mechanisms beyond the ones mentioned. This includes but is not limited to potentiometric sensors, optical sensors, electrochemical sensors, and surface plasmon resonance sensors. By incorporating these additional types of sensors, the authors can offer a more comprehensive evaluation of micro pH sensors' strengths, limitations, and potential applications relative to other sensing technologies. The following papers should be discussed. 10.1039/D0RA00016G, 10.1364/OE.511190, 10.3390/s21124218, 10.1016/j.snb.2023.133334.

2.     In the introduction section, the authors stated the importance of immobilizing a sufficiently thick organic layer for an optical pH sensor. Then, what are the comparative advantages of electrochemical grafting via cathodic reduction of aryl diazonium salts for immobilizing Neutral Red (NR) on metalized optical fibers versus stainless steel mirrors, in terms of control over layer thickness, chemical composition, and practical application in complex aqueous environments such as those anticipated in deep geological disposal facilities?

3.     In line 101, the ‘e’ should be capitalized.

4.     The section 2.2 is not clear to explain the experimental setup. How do two optical fibers combine? How does the mirror integrate with the fibers? Can authors provide microscope images of the fiber tip for both configurations?

5.     What are the key factors influencing the reproducibility and accuracy of electrochemical measurements on 316L stainless steel plates and mirrors, particularly regarding electrode choice and scanning speed, for optimizing the covalent grafting process and subsequent analytical characterizations?

6.     In section 3.3, what are the mechanisms responsible for the shift in the pH range of NR upon grafting to metalized surfaces, and how do these modifications influence its optical properties such as pKa values and isosbestic point wavelengths, as observed in UV-visible absorption spectra?

7.     Figure 7 is cropped. Full spectra need to be shown in the revision. Additionally, the explanation of Figure 7 is unclear. What do the authors mean by the wavelength of the isosbestic point shifting from 470 nm to 500 nm?

8.     The results in Figure 8 are not convincing. Although the trends of the three curves look similar, there is a difference in the Y-axis values. If the authors want to compare the absolute values, they need to normalize the data and then make the comparisons.

9.     In section 3.5, what are the effects of different radii of curvature and diameters of 316L stainless steel mirrors on the coupling efficiency of light in optical sensors, and how can these parameters be optimized for maximum efficiency?

10.  What are the effects of organic layer thickness on the UV-visible absorption sensitivity and selectivity of NR grafted on 316L stainless steel mirrors, and how can this relationship be optimized for pH sensing applications?

Author Response

Comment 1:   To provide a thorough analysis of micro pH sensors, authors should expand their comparison to encompass a broader range of sensing mechanisms beyond the ones mentioned. This includes but is not limited to potentiometric sensors, optical sensors, electrochemical sensors, and surface plasmon resonance sensors. By incorporating these additional types of sensors, the authors can offer a more comprehensive evaluation of micro pH sensors' strengths, limitations, and potential applications relative to other sensing technologies. The following papers should be discussed. 10.1039/D0RA00016G, 10.1364/OE.511190, 10.3390/s21124218, 10.1016/j.snb.2023.133334.

Response 1: Thank you for your review.

The following suggested articles find their application for pH measurement in biological media: 10.1039/D0RA00016G, 10.1364/OE.511190, 10.1016/j.snb.2023.133334. Unfortunately, we cannot cite these articles in our manuscript. Our optical pH sensor is used to monitor the pH of water in complex aqueous media. Please find in red some additional information (lines 54 to 56).

Comment 2 :  In the introduction section, the authors stated the importance of immobilizing a sufficiently thick organic layer for an optical pH sensor. Then, what are the comparative advantages of electrochemical grafting via cathodic reduction of aryl diazonium salts for immobilizing Neutral Red (NR) on metalized optical fibers versus stainless steel mirrors, in terms of control over layer thickness, chemical composition, and practical application in complex aqueous environments such as those anticipated in deep geological disposal facilities?

Response 2: Dear Rewiever, thank you so much for this question that you have kindly pointed out.

The 316L stainless steel mirror used has a diameter of 12.7 mm, while the metallized optical fiber has a core diameter of 940 μm.

Covalent grafting by cathodic reduction of diazonium aryl salts allows to graft a large amount of neutral red (NR) on the mirror, since its diameter is important.

The control of the layer thickness and the chemical composition are rigorously the same for the two types of surfaces (316L stainless steel mirror and metallized fiber). In a practical point of view, grafting onto the mirror surface avoids the metallization step in the case of fiber.

According to the literature, the grafting by cathodic reduction of diazonium aryl salts leads to the formation of an organic multilayer. We used diazonium aryl salt chemistry to create the covalent bonds between the chemical recognition phase and our conductive surfaces.  The major advantages of this electrochemical method under surface polarization are its fast to generate radicals in a gentle and progressive manner, observed on recorded voltammograms (figure 5 and figure 15). The thickness of the grafted chemical recognition phase is controlled through the concentration of diazonium salts used, the amount of charge consumed during the electro-grafting process, the number of grafting cycles and the scanning speed. The chemical composition of the recognition phase is controlled by the various analytical characterizations carried out, particularly the XPS (figure 16).

The practical application of optical pH probes in complex aqueous environments such as those anticipated in deep geological disposal facilities helps avoid chemical changes such as carbon dioxide partial pressure. In addition, the pH variation of pore water in geological formations is correlated with the existence chemical reactions involved in the degradation of clay, metals and concrete.

Comment 3 :       In line 101, the ‘e’ should be capitalized.

Response 3: Dear Rewiever, written in red please find the correction (line 109) as suggested.

Comment 4 : The section 2.2 is not clear to explain the experimental setup. How do two optical fibers combine? How does the mirror integrate with the fibers? Can authors provide microscope images of the fiber tip for both configurations?

Response 4 : Dear Rewiever, written in red please find additional explanation about two optical fibers combined as well as mirror integration (lines 117 to 119) as suggested. We provided microscope image of the fiber tip and mirror holder (lines 127, 129 and 130).

Comment 5 : What are the key factors influencing the reproducibility and accuracy of electrochemical measurements on 316L stainless steel plates and mirrors, particularly regarding electrode choice and scanning speed, for optimizing the covalent grafting process and subsequent analytical characterizations?

Response 5 : Dear Reviewer , thank you so much for this question.

Stainless steel 316L is an alloy. Charge transfer resistance, which determines the ease with which current can flow, is linked to its conductive nature and the existence of surface oxides. 316L stainless steel has a high charge transfer resistance due to its insulating nickel and chromium oxides. Electrode choice and scanning speed are key factors influencing the reproducibility and accuracy of electrochemical measurements.

Comment 6 :   In section 3.3, what are the mechanisms responsible for the shift in the pH range of NR upon grafting to metalized surfaces, and how do these modifications influence its optical properties such as pKa values and isosbestic point wavelengths, as observed in UV-visible absorption spectra?

Response 6 : Dear Reviewer, written in red please find the answer to this question that you have kindly to pointed out (lines 240 to 243)

Comment 7 :     Figure 7 is cropped. Full spectra need to be shown in the revision. Additionally, the explanation of Figure 7 is unclear. What do the authors mean by the wavelength of the isosbestic point shifting from 470 nm to 500 nm?

Response 7 : Dear Reviewer, please find full spectra for Figure 7 as suggested (line 274). Written in red please find sentences rephrased (lines 247 to 250).

Comment 8 : The results in Figure 8 are not convincing. Although the trends of the three curves look similar, there is a difference in the Y-axis values. If the authors want to compare the absolute values, they need to normalize the data and then make the comparisons.

Response 8 : Dear Reviewer, please find the spectra normalized for figure 8 and the explanation (lines 259 to 260).

Comment 9 :   In section 3.5, what are the effects of different radii of curvature and diameters of 316L stainless steel mirrors on the coupling efficiency of light in optical sensors, and how can these parameters be optimized for maximum efficiency?

 Response 9 : Dear Reviewer, written in red please find the answer to this question that you have kindly pointed out (lines 331 to 334).

Comment 10 :  What are the effects of organic layer thickness on the UV-visible absorption sensitivity and selectivity of NR grafted on 316L stainless steel mirrors, and how can this relationship be optimized for pH sensing applications ?

Response 10 : Dear Reviewer, think you so much for this question.

By grafting a thick organic layer onto the 316L stainless steel mirror, a usefull optical signal can be obtained, improving the sensitivity of the optical pH probe. Sensitivity can be also improved by optimizing electrochemical parameters, notably the number of grafting cycles and the scanning speed. The nature of the pH indicator molecule determines the selectivity.

Reviewer 2 Report

Comments and Suggestions for Authors

The authors present an interesting optical sensor for pH measurement using neutral red acid base indicator on the surface of metallized glass or metallic surfaces. In general, the methodology is clear, and the authors propose something new, it is sufficient for publication before some indications must be attended to.

Questions

1. It would be better if some more recent references were added, because most of the references are old (20-30 years old).

2. Lines 21-22: this sentence must be corrected because pH values above 7 are alkaline.

3. Lines 87-88: It would be appropriate to mention that the methodology will use the combination of covalent immobilization and electrochemical grafting, as it may imply that they were used separately.

4. Lines 82-86: add references that support these sentences.

5. Line 101: Please, put the capital letter at the beginning.

6. Line 243: To affirm that there are no significant differences, statistical analysis is required. For example, could be reported the coefficient of variation between the 3 replicates at a specific pH value (and at pH value where the absorbance is not higher than 1, since after this value, the absorbance readings are not reliable), to have an idea of how much the results varied.

 7. In figure 12, when the optical fiber is metallized with a CR/Au coating, is there any effect of surface plasmons on the measurements?

8. According to figure 14, where the metal mirror has a radius of curvature of 20 mm and 10 mm in diameter, do the authors use a collimated or point source? If they use the collimated source, does it mean that the concave surface of the metal mirror will converge the light beam in half the radius of curvature and thus obtain a light spot?

Author Response

Dear Reviewer, thank you so much for your kind and encouraging words regarding our work. It gives us driving force to continue pursuing ways to improve our optical pH sensors and its applications.

Comment 1: It would be better if some more recent references were added, because most of the references are old (20-30 years old).

Response 1: Thank you for your review. Please find in red some additional information (lines 54 to 56).

Comment 2 : Lines 21-22: this sentence must be corrected because pH values above 7 are alkaline.

Response 2 : Dear Reviewer, thank you so much for this comment. Written in red please find corrected sentence (lines 21 to 22).

Comment 3 : Lines 87-88: It would be appropriate to mention that the methodology will use the combination of covalent immobilization and electrochemical grafting, as it may imply that they were used separately.

Response 3 : Dear Reviewer , written in red please find the suggestion that you have kindly  pointed out (lines 91 to 93)

Comment 4 : Lines 82-86: add references that support these sentences.

Response 4  : Dear Reviewer, written in red please find added references as suggested (lines 86 and 88).

Comment 5 : Line 101: Please, put the capital letter at the beginning.

Response 5  : Dear Reviewer, written in red please find the correction (line 109) as suggested.

Comment 6 : Line 243: To affirm that there are no significant differences, statistical analysis is required. For example, could be reported the coefficient of variation between the 3 replicates at a specific pH value (and at pH value where the absorbance is not higher than 1, since after this value, the absorbance readings are not reliable), to have an idea of how much the results varied.

Response 6  : Dear Reviewer, please find the spectra normalized for figure 8 (line 277).

In order to compare the results, the spectra have been normalized as reported in Figure 8. Our first results obtained by reproducing the electrochemical grafting of NR on different metallized glass plates show that the Abs=f(pH) curves are almost superposable, which qualitatively shows that the process produces layers of fairly constant quality.

 Comment 7 :  In figure 12, when the optical fiber is metallized with a CR/Au coating, is there any effect of surface plasmons on the measurements?

Response 7 : Dear Reviewer, thank you for this comment. 

Our optical fiber is metallized with a deposit of 5 nm chromium and 25 nm gold. Although the deposited metal layers are semi-transparent, they are too thick to induce surface plasmon resonance effects. To our knowledge, there is no effect of surface plasmons on our measurements.

Comment 8 : According to figure 14, where the metal mirror has a radius of curvature of 20 mm and 10 mm in diameter, do the authors use a collimated or point source? If they use the collimated source, does it mean that the concave surface of the metal mirror will converge the light beam in half the radius of curvature and thus obtain a light spot?

Response 8 : Dear Reviewer, thank you for this comment. We used a point source.

Round 2

Reviewer 1 Report

Comments and Suggestions for Authors

The authors have made efforts to address my concerns in the revision; however, significant information is still missing, and several of my questions remain inadequately answered. 

Comment 1: The authors state that the proposed optical pH sensor is intended for monitoring the pH of water in complex aqueous media. However, the introduction contains numerous irrelevant references, such as 32-34, which are cited without clear justification. Furthermore, the paragraph in question is poorly structured, with excessive information that lacks a clear focus. The text shifts between different concepts without properly linking them, making it difficult for readers to follow the logical flow of the argument. While the paragraph discusses various immobilization methods and their associated literature, it fails to critically evaluate these methods. For instance, it mentions that adsorption and trapping methods lead to poor robustness but does not provide specific examples or explore the underlying reasons. Additionally, although many references are cited, the paragraph does not clarify what each reference contributes to the discussion, and the clustering of citations adds to the confusion about which studies support which points.

Comment 2: Although the authors have provided a detailed response to my question, the information should be integrated into the main manuscript rather than being confined to the response letter.

Comment 4: The authors did not adequately address my question regarding the integration of the mirror with the optical fibers. The explanation is unclear, and it remains confusing how the mirror is incorporated into the system. Additionally, images c and d are not microscope images of the fiber tip, as claimed. The manuscript also fails to specify the size of the mirror, which is essential information for understanding the experimental setup.

Comment 7: How can the absorption spectra go below 0?

New Comment: Figure 14 is too blurry to be legible. Section 3.5 should be thoroughly revised with higher-quality figures and more detailed results.

In conclusion, while the manuscript contains valuable information and the authors have made efforts to revise it, significant issues remain. These include missing information, logical inconsistencies, and technical errors. As a result, I cannot accept this manuscript for publication at this time.

Author Response

Comment 1: The authors state that the proposed optical pH sensor is intended for monitoring the pH of water in complex aqueous media. However, the introduction contains numerous irrelevant references, such as 32-34, which are cited without clear justification. Furthermore, the paragraph in question is poorly structured, with excessive information that lacks a clear focus. The text shifts between different concepts without properly linking them, making it difficult for readers to follow the logical flow of the argument. While the paragraph discusses various immobilization methods and their associated literature, it fails to critically evaluate these methods. For instance, it mentions that adsorption and trapping methods lead to poor robustness but does not provide specific examples or explore the underlying reasons. Additionally, although many references are cited, the paragraph does not clarify what each reference contributes to the discussion, and the clustering of citations adds to the confusion about which studies support which points.

Response 1 : Dear Reviewer, thank you so much for this comment. Written in red please find corrected paragraphs (lines 52 to 83).

Comment 2: Although the authors have provided a detailed response to my question, the information should be integrated into the main manuscript rather than being confined to the response letter.

Response 2: Dear Rewiever, thank you so much for this remark. Written in red please find the information integrated into the main manuscript (lines 106 to 116, lines 132 to 136 and lines 348 to 352).

Comment 4: The authors did not adequately address my question regarding the integration of the mirror with the optical fibers. The explanation is unclear, and it remains confusing how the mirror is incorporated into the system. Additionally, images c and d are not microscope images of the fiber tip, as claimed. The manuscript also fails to specify the size of the mirror, which is essential information for understanding the experimental setup.

Response 4 : Dear Rewiever, thank you so much for this comment. Please find in red information regarding the integration of the mirror with the optical fibers (lines 143 to 168, lines 170 to 172 and lines 174 to 175).

Comment 7: How can the absorption spectra go below 0?

Response 7: The absorption spectra are obtained by comparing two signals: metallized glass plate and metallized glass plate and electro-grafted with a 10 mM concentration of neutral red. The anomalies in the absorption values, such as values less than 0 or greater than 1, are due to the noise level suppression and the normalization process.

New Comment: Figure 14 is too blurry to be legible. Section 3.5 should be thoroughly revised with higher-quality figures and more detailed results.

New Response: Dear Rewiever, Written in red please find Section 3.5 revised (lines 365 to 380 and line 391).  

Round 3

Reviewer 1 Report

Comments and Suggestions for Authors

The authors have addressed my concerns.